# Structure and Physical Properties of Conductive Bamboo Fiber Bundle Fabricated by Magnetron Sputtering

**DOI:** 10.3390/ma16083154

**Published:** 2023-04-17

**Authors:** Wenqing Wang, Jiayao Li, Jiangtao Shi, Yue Jiao, Xinzhou Wang, Changlei Xia

**Affiliations:** 1Department of Wood Science and Engineering, College of Materials Science and Engineering, Nanjing Forestry University, Nanjing 210037, China; vntt@njfu.edu.cn (W.W.); lijiayao@njfu.edu.cn (J.L.); yjiao123@126.com (Y.J.); xzwang@njfu.edu.cn (X.W.); 2Co-Innovation Center of Efficient Processing and Utilization of Forest Resources, Nanjing Forestry University, Nanjing 210037, China

**Keywords:** conductive fiber, magnetron sputtering, bamboo

## Abstract

The variety of conductive fibers has been constantly enriched in recent years, and it has made rapid development in the fields of electronic textiles, intelligent wearable, and medical care. However, the environmental damage caused by the use of large quantities of synthetic fibers cannot be ignored, and there is little research on conductive fibers in the field of bamboo, a green and sustainable material. In this work, we used the alkaline sodium sulfite method to remove lignin from bamboo, prepared a conductive bamboo fiber bundle by coating a copper film on single bamboo fiber bundles using DC magnetron sputtering, and analyzed its structure and physical properties under different process parameters, finding the most suitable preparation condition that combines cost and performance. The results of the scanning electron microscope show that the coverage of copper film can be improved by increasing the sputtering power and prolonging the sputtering time. The resistivity of the conductive bamboo fiber bundle decreased with the increase of the sputtering power and sputtering time, up to 0.22 Ω·mm; at the same time, the tensile strength of the conductive bamboo fiber bundle continuously decreased to 375.6 MPa. According to the X-ray diffraction results, Cu in the copper film on the surface of the conductive bamboo fiber bundle shows the preferred orientation of (111) the crystal plane, indicating that the prepared Cu film has high crystallinity and good film quality. X-ray photoelectron spectroscopy results show that Cu in the copper film exists in the form of Cu^0^ and Cu^2+^, and most are Cu^0^. Overall, the development of the conductive bamboo fiber bundle provides a research basis for the development of conductive fibers in a natural renewable direction.

## 1. Introduction

With the popularization of the concept of natural environmental protection, the whole manufacturing industry needs to transform to green and environmental protection. The traditional fiber reinforced polymer (FRP) composites, typically using synthetic fibers, such as aramid, glass fiber, and carbon fiber, owing to their high strength and reliability [1], are currently faced with the challenges of economic cost-effectiveness and ecological environmental friendliness; therefore, finding a fully biodegradable and low-cost fiber-reinforced polymer composite becomes the key to the problem [2,3]. Among them, cellulose-rich bamboo has attracted a lot of attention from researchers in this field.

Bamboo, a woody fibrous material with an organic composition as the main chemical component, is a natural polymer mainly composed of cellulose, lignin, and hemicellulose, which play the role of skeleton, shell, and matrix, respectively. In last 40 years, a variety of bamboo composites have been developed [4]. Bamboo resources not only have superior physical and mechanical properties, are strong and durable, and are easy to produce and process, but also have a short growth cycle as well as being a sustainable and natural renewable material. However, compared to wood, the round, hollow geometry and structural heterogeneity of bamboo make the processing costly and inefficient [5]. Therefore, the preparation of FRP after separating bamboo into bamboo fibers has become a research hot spot. Bamboo fiber bundles with chemical treatment to remove lignin have very strong mechanical properties, much higher than natural bamboo fibers, and are even comparable to some carbon fibers [6]. As such, the use of delignified bamboo fibers, such as the FRP composite, can effectively improve various mechanical properties. Some researchers found that the tensile and flexural properties of the fiber-reinforced composites prepared by embedding bamboo fibers in thermosetting resins were proportional to the volume admixture of bamboo fibers (0–40%), and the strength of the FRP composite at higher fiber loading was dominated by the bamboo fiber strength [1].

The advancement of electronic informatization has sparked a strong interest in conductive fibers owing to their high conductivity, portability, and durability. There are many preparation methods for conductive fibers, and they can be divided into three types of conductive fibers, including: metal-based, carbon-based, and polymer-based [7]. Obviously, the conductive fibers exhibit excellent chemical and physical properties, and researchers are now moving toward the soft, deformable, breathable, durable, portable, and practical conductive fibers. The mechanical properties of these fibers are usually excellent, with a typical Young’s modulus ranging from a few MPas to KPas. They are flexible and pliable and can deform under small external forces or their own weight, and they typically have a large specific surface area of about 10^2−3^ m^2^/kg, while they can also be stretched, twisted, bent, or sheared for different applications [8]. The preparation strategies of conductive fibers can be divided into two main types: one is the use of conductive materials, such as polymers, to form solutions or melts for coblending; the other is the physical and chemical deposition and coating of conductive substances on (or within) the fiber surface to form a thin film [7], mainly including chemical plating [9], electroplating [10], chemical vapor deposition (CVD) [11], physical vapor deposition (PVD) [12], and so on. Magnetron sputtering is one of the methods of PVD. The film prepared by magnetron sputtering has a higher density and purity, a smoother surface, and a higher adhesion between the film and the substrate [13]. It has been widely used in the preparation of conductive fibers.

For conductive fibers, electric conductivity is an important physical property, especially for conductive fibers used on wearable devices. Usually, we use conductivity or resistivity to characterize the electric conductivity of the sample. In this study, resistivity was selected to characterize the electric conductivity of the conductive bamboo fiber bundle. The resistivity (ρ) is calculated as (1) (resistance (R), area (A), and length (L)) [14]:R = ρL/A(1)

There are some studies on conductive bamboo fibers. Gao et al. [15] reported a multifunctional bamboo preparation method with superhydrophobic, self-cleaning, conductive, flame-retardant, and antibacterial properties. Ag NPs were loaded in situ on the surface of bamboo through an electroless Ag plating reaction, and, finally, 1H,1H,2H,2H-perfluorodecanethiol (PFDT) deposition was carried out on the surface. Lin et al. [16] proposed a one-step alkaline pretreatment method to prepare superconducting BCFs by the in situ reduction of Ag NPs. The pretreated BCFs achieve an in situ alkali trigger for depositing and uniting Ag NPs during the electroless Ag plating process. These conductive bamboo fibers have good electrical conductivity and mechanical strength; however, magnetron sputtering, as an emerging coating technology, has not been used in the preparation of conductive bamboo fibers.

In this work, after lignin removal by the alkaline sodium sulfite method, the conductive bamboo fiber bundle was prepared by coating metallic copper onto the surface of the bamboo fiber bundle using the DC magnetron sputtering technique. Compared with traditional conductive fibers, the raw materials used in conductive bamboo fiber bundles are greener and have great conductive and mechanical properties. The effect of process parameters on their structural and physical properties was analyzed using morphology study, resistivity analysis, tensile strength analysis, X-ray diffraction (XRD) analysis, and X-ray photoelectron spectroscopy (XPS) analysis. Finally, the conductive bamboo fiber bundle with an integrated cost and performance were prepared, which provides a theoretical basis for the subsequent development and utilization of conductive bamboo fiber bundles in the future, and covers part of the gap in the application of magnetron sputtering in the field of wood and bamboo materials.

## 2. Materials and Methods

### 2.1. Raw Materials

The natural moso bamboo (3–5 years of age) was collected from Nanjing Forestry University, Nanjing, Jiangsu Province. Hydrogen peroxide, sodium hydroxide, and anhydrous sodium sulfite were purchased from Nanjing Chemical Reagent Co., Ltd. (Nanjing, China), while deionized water (DI) was prepared in the laboratory. Copper target (99.995% purity) was purchased from ZhongNuo Advanced Material (Beijing) Technology Co., Ltd. (Beijing, China).

### 2.2. Fiber Treatment

The natural bamboo was cut into slices of 20 cm long (excluding joints), the bamboo bark of the surface was scraped off and boiled for 1 h to remove water-soluble organic substances and trapped air. Subsequently, the treated bamboo pieces were impregnated in a mixed solution of 4 mol/L NaOH and 2.5 mol/L Na_2_SO_3_, lignin removal was conducted at 80 °C for 6 to 8 h, then bleached with 30% H_2_O_2_ and washed with deionized water several times to remove the chemicals. Finally, the bamboo was placed in the oven at 60 °C for 6 to 8 h [17], and the treated bamboo pieces were divided into a single bamboo fiber bundle with a length of 60 mm and a diameter of about 0.3 mm.

### 2.3. Preparation of Conductive Bamboo Fiber Bundle

Ten bamboo fiber bundles were divided into one group and put into the vacuum chamber of the high-vacuum magnetron sputtering film deposition system (TRP-450). Ar was selected as the sputtering gas and the gas flow rate was set at 60 Sccm. Presputtering was done for 60 s before sputtering to improve the stability of the sputter coating. The sputter pressure was set at 3.0 Pa and the rotation speed of the substrate turntable was controlled at 5 Rpm. Each group of bamboo fiber bundles was sputtered on both sides to ensure that the copper thin film was evenly deposited on the surface of the bamboo fiber bundles. Lower DC magnetron power results in lower sputtering rates and also leads to lower film deposition rates [18], so we set the sputtering power as 100 w, 150 w, and the sputtering time as 600 s, 900 s, and 1200 s [19,20], which were divided into six groups and labeled as 100-6, 100-9, 100-12, 150-6, 150-9, and 150-12, respectively. A single conductive bamboo fiber bundle has a length of 60 mm, a diameter of 0.3 mm, and a density of 0.82 g/cm^3^. The density of the conductive bamboo fiber bundle was not affected by the magnetron. The bamboo fiber bundle without the magnetron sputtering treatment was named as the CK group.

## 3. Methodology

The morphologies of the conductive bamboo fiber bundle were observed utilizing an optical microscope (OLYMPUS BX51, Olympus, Singapore) and scanning electron microscope (SEM, TM-1000, Palangka Raya, Indonesia). The resistivity was measured with a digital multimeter (VICTOR VC890D, Bei Cheng (Hong Kong) Technology Co., Limited, (Shenzhen, China). In order to explore the conductivity differences of the conductive bamboo fiber bundle under different ambient temperatures, samples with a sputtering power of 150 w and sputtering time of 900 s were selected and sent to an oven at 45 °C and a refrigerator at −2 °C to simulate the high temperature in summer and low temperature in winter, which was then processed for 24 h and 48 h, respectively, and the change of resistivity was recorded. The mechanical tensile properties were evaluated using a universal mechanical testing machine (CMT4204, MTS (China), Shanghai, China) for each treatment group, and five samples were tested with a tensile speed of 5 mm/min. XRD (Ultima IV, Rigaku, Tokyo, Japan) and XPS (AXIS UltraDLD, Kratos Analytical, Manchester, UK) were used to investigate the chemical compositions of the samples. XRD was recorded from 2*θ* = 10–80° and the test speed was 8°/min, while the XPS instruments were equipped with an Al-Kα source (hν = 1486.6 eV) as the X-ray source with 600 w power.

## 4. Results and Discussion

### 4.1. Morphology Study of Conductive Bamboo Fiber Bundle

Figure 1a shows that the surface of 150-9 covered with a layer of purplish red film compared to the CK. Since no other substances were introduced in the experiment, it can be basically determined that the film covered is copper film. After delignification, pits and fiber gaps can be seen from the SEM images of the bamboo fiber bundles (Figure 1b). When the sputtering power was 100 w and the sputtering time was 600 s, the copper film covering the fiber surface was uneven, with poor coverage and a large number of cracks (Figure 1c). At the same sputtering time, and with the sputtering power increased to 150 w, the number of cracks was significantly reduced, but the copper film remained uneven and the uniformity and continuity were not greatly improved (Figure 1d). When the sputtering time was extended to 900 s, the copper film covering the surface of the two groups of samples with a sputtering power of 100 w and 150 w was continuous without obvious cracks, but there were still some defects affecting the performance of conductive bamboo fiber bundle (Figure 1e,f). Moreover, when the sputtering time reached 1200 s, the copper film covering effect on the surface of both groups of samples was better, there were no obvious defects, and the uniformity and continuity were greatly improved (Figure 1g,h). Therefore, it can be concluded that the sputtering time has more influence on the uniformity and continuity of the sputtered film compared with the sputtering power. At a certain sputtering power, the covering effect of the sputtered film becomes better with the increasing sputtering time, and the uniformity and continuity are improved, while the sputtering power has less influence on the sputtered film.

### 4.2. Resistivity Analysis

As can be seen from Figure 2a, when the sputtering time was 600 s, the resistivity of the sample with the 150 w sputtering power was 0.73 Ω·mm, and compared with that of the sample with the 100 w sputtering power, the resistivity reached 3.51 Ω·mm, which is close to five times that of the 150 w group. When the sputtering time was extended to 900 s, compared with 600 s, the resistivity of both groups showed a large decrease of 56.3% and 57.7%, indicating that the electrical conductivity of both groups was greatly improved. When the sputtering time reached 1200 s, compared with 900 s, the amount of resistivity decrease of the two groups was significantly reduced, with the resistivity of the 150 w group decreasing by 27.5% while the resistivity of the 100 w group decreased by only 6%. Combined with the results of the SEM analysis, when the sputtering time was 600 s, the covering effect of the two groups of samples was poor and failed to form a complete and continuous copper film, and the resistivity of the conductive bamboo fiber bundle was higher at this time. When the sputtering time was 900 s, the surface of the fiber bundle was covered with a complete copper film, and the thickness of copper film became thicker, so the electrical conductivity of both groups of samples was improved to a larger extent. Additionally, when the sputtering time was extended to 1200 s, the resistivity of the conductive bamboo fiber bundle continued to decrease as the thickness of the copper film increased. Although the subsequent copper particles were continuously deposited onto the sample surface, the effect on the continuity and uniformity of the copper film was diminished; therefore, the change in the resistivity of the conductive bamboo fiber bundle was decreased. Hiba et al. [21] explored the relationship between the copper thickness and sheet resistance and found that, when the thickness of the copper increased from 5 nm to 15 nm to 20 nm, the sheet resistance of ZnO/Cu/ZnO multilayer film was reduced from 95.1 to about 22.5 Ω/sq, and finally decreased sharply to 9.877 Ω/sq, respectively, which is similar to the results of this study. It can be considered that the sputtering power has a great influence on the conductivity of the conductive bamboo fiber bundle, as the resistivity decreased significantly when the sputtering power increased at a certain sputtering time. The sputtering time also had an effect on the conductivity, and the resistivity decreased more significantly when the sputtering time was extended from 600 s to 900 s, and less when it was extended from 900 s to 1200 s.

Figure 2b shows the resistivity difference graph of the conductive bamboo fiber bundles at different ambient temperatures. The resistivity of the fiber bundle increased by 31.7% and 46.2% after 24 h and 48 h drying treatments, respectively. This may be because the high temperature caused the evaporation of water from the fiber bundle and the resistivity of the copper film on the surface increased, resulting in a certain degree of reduction of the electrical properties of the fiber bundle. However, the resistivity of the fiber bundles only increased by 6.1% and 7.9% after 24 h and 48 h freezing treatments, respectively, indicating that low temperature has little effect on the resistivity of the fiber bundle and hardly affects its performance.

We compared the resistivity of the conductive bamboo fiber bundles with some currently reported conductive fibers (Figure 2c), including conductive fibers for biomedical applications and wearable conductive fibers [15,16,22,23,24,25,26,27,28,29,30,31], and found that conductive bamboo fiber bundles have superior electrical conductivity and are comparable to some wearable conductive fibers.

### 4.3. Tensile Strength Analysis

The separated bamboo fiber bundle has thick cell walls assembled by aligned semicrystalline lignocellulosic microfibrils, demonstrating an outstanding tensile strength [6]. We evaluated the tensile mechanical properties of the separated bamboo fiber bundle (Figure 3a), where the tensile strength of bamboo fiber bundle (CK) without magnetron sputtering treatment was 858 MPa, while the tensile strength of several groups of samples after magnetron sputtering decreased significantly (44–56%), and the tensile strength kept decreasing with the extension of the sputtering time and the improvement of sputtering power [32]. When the sputtering power was 150 w and the sputtering time was 1200 s, the tensile strength decreased to the lowest value of 375.6 MPa. The covering of copper film would not destroy the structure of the bamboo fiber bundle, so the reason for the reduction of the tensile strength is the covering of copper film. The tensile strength of Cu was about 209 MPa, which is only 1/4 of that of the untreated bamboo fiber bundle. The copper film part would break before the bamboo fiber bundle broke; therefore, the addition of the Cu film would definitely lead to the reduction of tensile strength. Due to the weakening of the bonding of the grains, copper segregation occurred in the grain boundary region around the fiber bundle, the stress of the conductive fiber bundle decreased significantly, and, with a further addition of copper, the number of copper grains increased accordingly. The mechanical properties of the conductive bamboo fiber bundle were greatly affected, which is consistent with the previous findings [33].

We compared the tensile strength of our fiber with a few reported cellulose-based macrofibres, carbon fiber composites, and conductive fiber [9,16,20,31,34,35,36,37,38,39] (Figure 3b), and found that the tensile strength of the bamboo fiber bundle was better than that of some other conductive fibers, and was comparable to cellulose-based macrofibres and carbon fiber composites. This shows that though the tensile strength of the bamboo fiber bundle was greatly reduced after covering with the copper film and still remained at a high level in the fiber materials.

### 4.4. XRD Analysis

Figure 4 shows the XRD spectra of the CK group and the other six groups of samples. Cellulose is one of the main components of bamboo, and therefore, it can be seen that seven curves show distinct diffraction peaks at 34.58°, 22.12°, and 15.72°, corresponding to three typical peaks in the (040), (002), and (101) lattice planes of cellulose I. Among them, the peak in the (002) plane reflects the width of the crystallization zone, the region between peaks (101) and (002) represents the amorphous components of cellulose, while the peak in the (040) plane represents the length of the crystallization zone [40,41]. Three diffraction peaks of Cu (111), Cu (200), and Cu (220) can be seen in six curves, except the CK group in Figure 4, signifying the polycrystallinity of the Cu film. The strong diffraction peaks observed at 2θ = 43° were attributed to Cu (111), suggesting a preferential orientation in the (111) direction [42]. With the increase of sputtering power and the extension of sputtering time, the preferential orientation did not change. Cu has a face-centered cubic (FCC) crystal structure, and for this structure, the (111) face is the most stable configuration, which has the lowest surface energy [43]; therefore, the prepared Cu film has high crystallinity and good film quality. Moreover, by increasing sputtering power and prolonging the sputtering time, the intensity of the three diffraction characteristic peaks were improved, indicating that the quality of copper film was improved and manifested as a decrease in resistivity.

### 4.5. XPS Analysis

As shown in Figure 5, we analyzed the surface elemental composition and chemical states of the CK group and six groups of samples after magnetron sputtering using XPS. Similar to wood, the main chemical composition of bamboo is organic, mainly composed of cellulose (about 55%), lignin (about 25%), and hemicellulose (about 20%), and the major components of bamboo include carbon, hydrogen, and oxygen [44], though hydrogen cannot be detected by XPS. It can be seen from Figure 5a that the six groups of samples, except the CK group, have obvious Cu 2p peaks. The appearance of the C 1s and O 1s peaks is due to the element composition of the bamboo fiber bundle itself. The high-resolution XPS spectrum of copper 2p is shown in Figure 5b. No predominant peak was found in the CK group within this range, and the predominant peaks of the other six groups of samples appear at 932.1 eV and 951.7 eV, which could be attributed to the spin–orbit (2p_3/2_ and 2p_1/2_) of Cu^0^ [45], and the sharp peak at about 932.1 eV, signifying the formation of a pure metallic copper film on the surface of the bamboo fiber bundle [42]. Moreover, Cu^0^ cannot be distinguished from Cu^+^ by XPS because their peaks of 2p_3/2_ and 2p_1/2_ are almost at the same location [46]; however, during the preparation of the conductive bamboo fiber bundles, the conditions for the formation of Cu^+^ were not met, so it can be considered that these two obvious peaks belong to Cu^0^. The insignificant peaks at 934.3 eV and 954 eV are 2p3/2 and 2p1/2 of Cu^2+^, and the presence of Cu^2+^ can be further confirmed by the two satellite peaks at 943.1 eV and 963 eV [47]. The appearance of Cu^2+^ may be due to the oxidation of Cu on the surface of the conductive bamboo fiber bundle, which reacts with O in the air to produce copper oxidation. Moreover, the peaks at 934.3 eV and 954 eV are not obvious or even visible for some samples, as seen in Figure 5b. This proves that most of the Cu elements of the copper film are in the form of Cu^0^.

## 5. Conclusions

In summary, we have prepared a cellulose-based, sustainable, and biodegradable conductive bamboo fiber bundle obtained by depositing copper film on the surface of a bamboo fiber bundle after chemical delignification, and demonstrated good electrical conductivity and tensile strength. As seen from the SEM images, when the sputtering power is 150 w and the sputtering time is 900 s, the copper film on the surface of the bamboo fiber bundle has basically covered the surface of the bamboo fiber bundle, and the coverage effect continues to improve with the increase of the sputtering power and sputtering time. The conductive bamboo fiber bundles have good electrical conductivity, and at a sputtering power of 150 w and a sputtering time of 900 s, the resistivity of the conductive bamboo fiber bundle was only 0.31 Ω·mm, while the effect of the sputtering time on the resistivity was small compared to the sputtering power. After coating the bamboo fiber bundles, the tensile strength of the fiber bundles showed a relatively large decrease (44–56%) and kept decreasing slightly with the increasing sputtering time and sputtering power, but remained at a high level. From the XRD results, the copper film on the surface of the conductive bamboo fiber bundle is highly crystallized and the film formation quality is good. According to the XPS results, Cu in the copper film exists in the form of Cu^0^ and Cu^2+^, and most of them are Cu^0^. Overall, in this experiment, a conductive bamboo fiber bundle with excellent electrical and mechanical properties was prepared. When the sputtering power was increased to 150 w and the sputtering time was extended to 900 s, the film quality and electrical conductivity of the copper film were greatly improved, and the performance of the conductive bamboo fiber bundle was not greatly improved when the sputtering time was extended to 1200 s. At the same time, the tensile strength of the fibers did not change significantly with the increase of sputtering power and time. Combining cost and performance, when the sputtering power is 150 w and sputtering time is 900 s, the prepared conductive bamboo fiber bundle is most suitable. The conductive bamboo fiber bundle proposed in this study could be used in light-weight portable electronics and wearable devices, providing a new way for the high value-added utilization of natural bamboo, and the preparation method provides a new idea for the preparation of conductive bamboo fiber.

## Figures and Tables

**Figure 1 materials-16-03154-f001:**
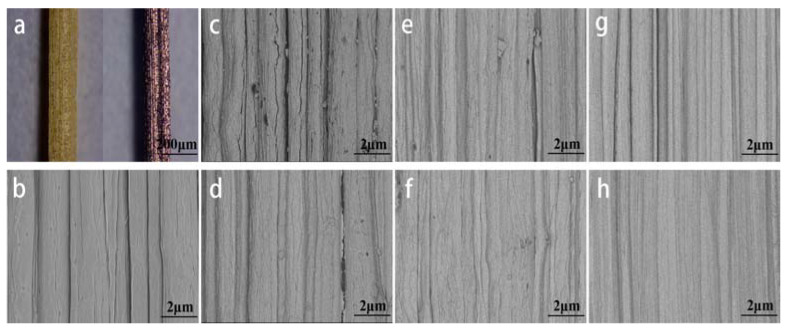
Microscopic picture of bamboo fiber bundle. (**a**) Stereomicroscope picture of CK and 150-9. (**b**–**h**) SEM images of CK and samples after magnetron sputtering: (**b**) CK, (**c**) 100-6, (**d**) 150-6, (**e**) 100-9, (**f**) 150-9, (**g**) 100-12, (**h**) 150-12.

**Figure 2 materials-16-03154-f002:**
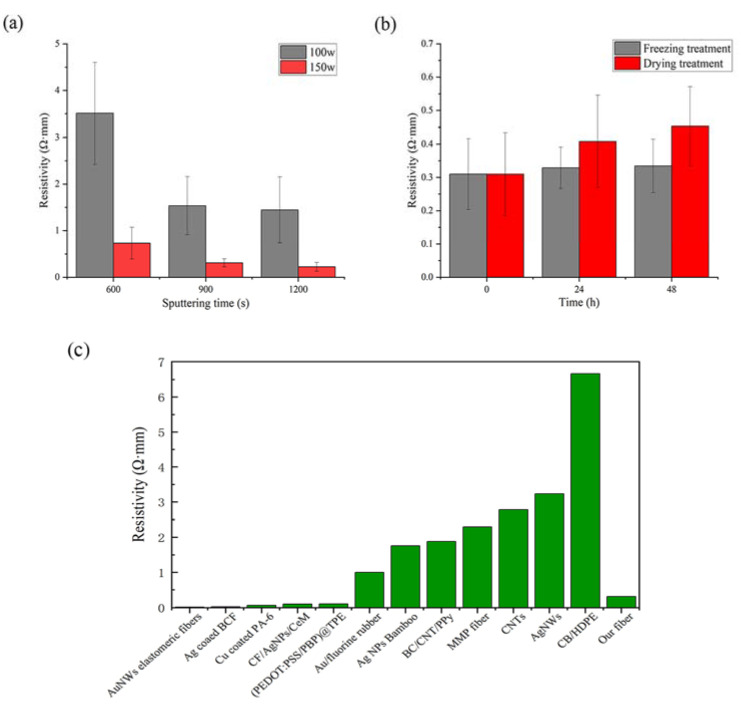
(**a**) Resistivity of conductive bamboo fiber bundle under different process parameters. (**b**) Resistivity of conductive bamboo fiber under different ambient temperatures. (**c**) Comparison of the resistivity of the conductive bamboo fiber bundle with other conductive fibers.

**Figure 3 materials-16-03154-f003:**
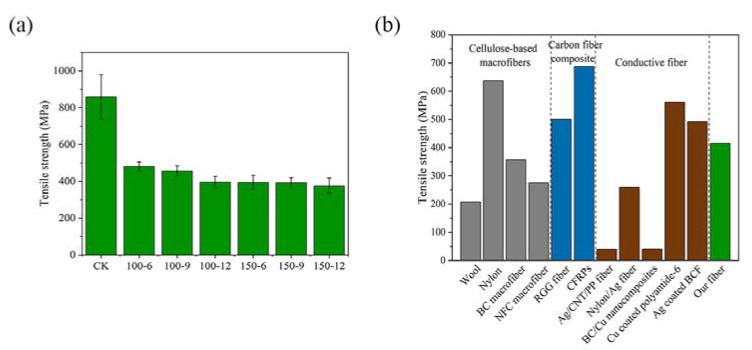
(**a**) The tensile mechanical properties of the bamboo fiber bundle and samples after magnetron sputtering. (**b**) Comparison of tensile strength between conductive bamboo fiber bundle and other fiber materials.

**Figure 4 materials-16-03154-f004:**
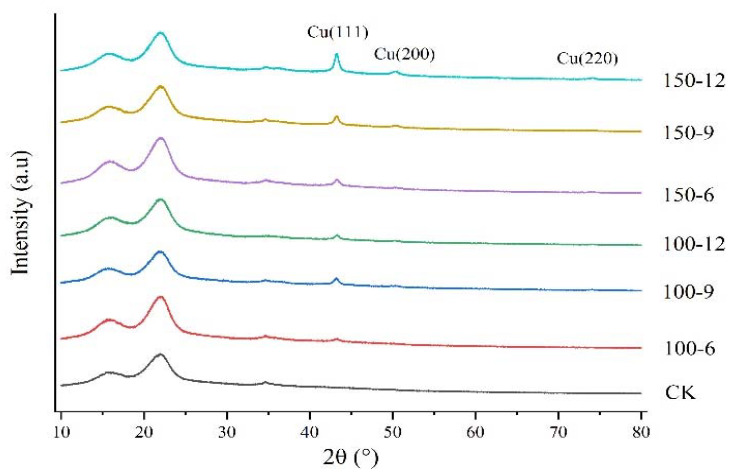
XRD patterns of the bamboo fiber bundle and samples after magnetron sputtering.

**Figure 5 materials-16-03154-f005:**
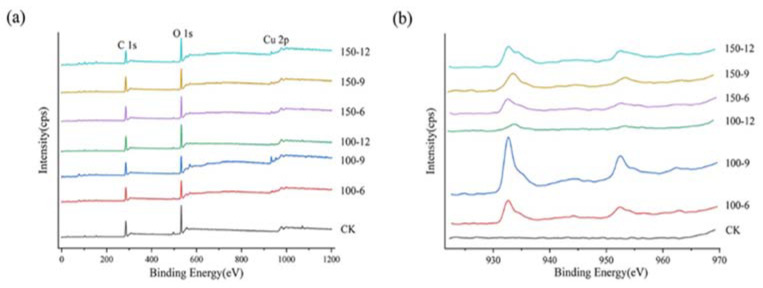
(**a**) The complete XPS spectra of the bamboo fiber bundle and samples after magnetron sputtering. (**b**) The high-resolution XPS spectrum of copper 2p.

## Data Availability

The data presented in this study are available on request from the corresponding author.

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
