# Peer review of "Structure and Physical Properties of Conductive Bamboo Fiber Bundle Fabricated by Magnetron Sputtering"

_materials, 2023, doi:10.3390/ma16083154_

Round 1

Reviewer 2 Report

Structure and physical properties of conductive bamboo fiber 2 bundle fabricated by magnetron sputtering

Emad's review points

In this study, a sustainable and biodegradable conductive bamboo fiber bundle have been prepared. Good electrical conductivity and tensile strength has been obtained by depositing copper film on the surface of bamboo fiber bundle after chemical delignification The title matches the content, and excellent work has been presented. I recommend the publication of the manuscript after the following improvements is done.

1.     Very long paragraph, and it is difficult to follow the main idea of the paragraph (line 140-163)

2.     The conclusion is very brief and does not provide suggestion for the next step (line 254-274).

Reviewer 4 Report

The presents an innovative idea to coat bamboo fibers using sputtering technique for making conductive fiber. However some improvement is required based on the following comments.

1. Current state of the art needs to be reviewed in the introduction

2. Clearly articulate the research gap and novelty of your work

3. Further details about XRD and XPS characterisation procedure needs to be included

4. Some idea of coating thickness should be provided to achieve the best conductivity

5. Explanation on why the resistivity decreased with the increase of sputtering power and time is required. is this linked with coating thickness?

6. Similarly no explanation has been offered on why the tensile strength decreased with sputtering power and time.

7. Conclusion on excellent mechanical properties and cost effectiveness have not been justified

Round 2

Reviewer 3 Report

Incorporated all the suggestion given earlier.

Reviewer 4 Report

the corrections are made according to the suggestions